# Survival of Visual Function in Patients with Advanced Glaucoma after Standard Guarded Trabeculectomy with MMC

**DOI:** 10.3390/jcm12041639

**Published:** 2023-02-18

**Authors:** Theodoros Filippopoulos, Dimitrios Tsoukanas, Stylianos A. Kandarakis, Angeliki Salonikiou, Michalis Georgiou, Fotis Topouzis

**Affiliations:** 1Athens Vision Eye Institute, Kallithea, 17673 Athens, Greece; 2First Department of Ophthalmology, G. Gennimatas Hospital National and Kapodistrian University of Athens, 11527 Athens, Greece; 3First Department of Ophthalmology, AHEPA Hospital, Aristotle University of Thessaloniki, 54636 Thessaloniki, Greece; 4Moorfields Eye Hospital, London EC1V 2PD, UK; 5UCL Institute of Ophthalmology, University College London, London EC1V 9EL, UK; 6Jones Eye Institute, University of Arkansas for Medical Sciences, Little Rock, AR 72205, USA

**Keywords:** guarded trabeculectomy, glaucoma surgery, end-stage glaucoma, advanced glaucoma, wipe-out, complications, loss of vision, survival of visual function

## Abstract

Surgical intervention in patients with severe glaucoma remains controversial, especially in unilateral cases with a minimally affected fellow eye. Many question the benefit of trabeculectomy in such cases due to high complication rates and prolonged recovery. In this retrospective, non-comparative, interventional case series we aimed to determine the effect of trabeculectomy or combined phaco-trabeculectomy on the visual function of advanced glaucoma patients. Consecutive cases with perimetric mean deviation loss worse than −20 dB were included. Survival of visual function according to five predetermined visual acuity and perimetric criteria was set as the primary outcome. Qualified surgical success utilizing two different sets of criteria commonly used in the literature constituted secondary outcomes. Forty eyes with average baseline visual field mean deviation −26.3 ± 4.1 dB were identified. The average pre-operative intraocular pressure was 26.5 ± 11.4 mmHg and decreased to 11.4 ± 4.0 mmHg (*p* < 0.001) after an average follow-up of 23.3 ± 15.5 months. Visual function was preserved at two years in 77% or 66% of eyes respectively according to two different sets of visual acuity and perimetric criteria. Qualified surgical success was 89%, 72% at 1 and 3 years respectively. Trabeculectomy and/or phaco-trabeculectomy is associated with meaningful visual outcomes in patients with uncontrolled advanced glaucoma.

## 1. Introduction

Glaucoma is a progressive optic neuropathy that affects visual function by initially restricting peripheral vision and eventually leading to central visual loss in late stages [1]. Glaucoma is currently a leading cause of irreversible blindness in the western world [2] and can have devastating effects on many aspects of everyday life with progressive disease being directly linked to quality of life [1]. It is also associated with notable management costs for healthcare systems [3]. Intraocular pressure (IOP) is the only modifiable risk factor for glaucoma to date and its reduction remains the mainstay of effectively treating glaucoma and delaying vision loss [4,5].

Standard guarded trabeculectomy with antimetabolites remains the most commonly employed surgical option for patients with severe glaucoma who are not controlled on maximal topical therapy and/or are progressing [6,7,8], despite visual acuity often declining after surgery as a result of the disease’s natural history, the frequency and magnitude of surgical complications and cataract progression [9,10,11]. Furthermore, loss of neuroretinal tissue can be slowed down but cannot be halted completely due to age-related retinal ganglion cell (RGC) loss [12]. Quality of life is related to the degree of binocular visual field loss [13,14]. As a result, in patients with asymmetric unilaterally severe glaucoma the question often arises, to what degree a surgical intervention is warranted, especially in cases with limited anticipated risk for the fellow minimally affected eye. Many practitioners and/or patients may be hesitant to proceed with surgery in such cases due to high complication rates [11], prolonged recovery and/or fear of the wipe-out phenomenon [15,16] or they may feel that an intervention is essentially non-beneficial as patients may continue to lose vision despite an otherwise successful intervention resulting in low IOP. Furthermore, delayed treatment requires a significantly lower rate of progression to effectively prevent further functional impairment which translates into lower target IOP [17] and narrower therapeutic window.

The purpose of this retrospective study was to determine survival of visual function utilizing both visual acuity and perimetric criteria in patients with severe glaucoma who underwent standard guarded trabeculectomy or combined phaco-trabeculectomy for medically uncontrolled glaucoma at two tertiary glaucoma referral centers by two fellowship-trained glaucoma surgeons.

## 2. Materials and Methods

### 2.1. Patient Selection

This is a retrospective, non-comparative, interventional case series of 40 consecutive patients/eyes with advanced glaucoma operated between January 2010 and December 2011 by two surgeons at two tertiary centers (TF, Athens Vision Eye Institute and FT, Ophthalmology Department of the Aristotle University of Thessaloniki at the AHEPA Hospital). Patients underwent either stand-alone standard guarded trabeculectomy or trabeculectomy combined with phacoemulsification and intraocular lens (IOL) implantation (phaco-trabeculectomy). Institutional review board approval was obtained to permit review of surgical log books at both institutions for this retrospective study. The research adhered to the tenets of the Declaration of Helsinki and was approved by the Human Research Ethics Committee of the Athens Vision Eye Institute and the local Ethics Committee in Thessaloniki. Informed consent was obtained from all patients prior to surgery explaining indications and risks of the procedure.

Included were adult patients with severe glaucoma, such as Primary Angle Glaucoma (POAG) or Exfoliative Glaucoma (XFG), not controlled and/or progressing on maximal medical topical or systemic treatment that underwent standard guarded trabeculectomy with mitomycin C (MMC) or phaco-trabeculectomy with MMC. Severe glaucoma was defined as visual field loss consistent with glaucoma with a mean deviation (MD) on standard automated perimetry (SAP) < −20.01 dB at least on two separate occasions prior to surgery, both of which demonstrated reliable visual field indices (fixation losses < 20%, false positive and false negative rate < 33%). The Humphrey Visual Field Analyzer (HFA II-i, Carl Zeiss Meditec Inc., Dublin, CA, USA) was utilized for visual field testing and the 24/2 SITA-Standard protocol was used in all instances. All patients had at least one visual field test on file in the 2 months preceding surgery. In cases where both eyes were eligible for analysis, one eye was randomly selected and included in our analysis utilizing an online randomization tool (www.random.org).

Exclusion criteria included previous incisional glaucoma surgery and intraoperative complications during cataract surgery such as posterior capsular rupture with vitreous prolapse and/or IOL implantation outside the capsular bag. Additional exclusion criteria were short or inconsistent follow-up (follow-up less than 6 months unless patients failed prior to that), inability to perform SITA-Standard Automated Perimetry with size III stimulus size reliably and pre-operative best corrected Snellen visual acuity (BCVA) < 20/200.

### 2.2. Clinical Data

For this retrospective study, pre-operative data collected from the patients’ charts included age, race, gender, diagnosis, study eye, BCVA, number and type of glaucoma medications, IOP, global visual field indices, central corneal thickness (CCT), cup to disk ratio, number and type of previous intraocular surgeries and previous glaucoma laser procedures. A substantial proportion of patients (80%) had pre-op split fixation documented on SAP. Split Fixation was defined as at least one out of four innermost central points on 24/2 SAP depressed at a level of *p* < 0.05 in the pattern standard deviation plot [18]. Intraoperative data included intraoperative complications, duration of MMC application and type of procedure (stand-alone trabeculectomy or phaco-trabeculectomy).

Follow-up data had been typically recorded on day one, day three, week one, week two and at 1, 3, 6, 12 months postoperatively and every 6 months thereafter and included IOP, glaucoma medication requirements, BCVA, early and late complications related to surgery as well as further surgical interventions. Post-op data such as IOP, BCVA and medication requirements were censored at the time of re-operation for glaucoma and not at the time of failure according to functional criteria. Additional post-op visits and/or interventions at the slit lamp such as 5-fluorouracil (5-FU) injections, bleb needlings and laser suture lyses had been at the discretion of the treating physician, did not classify as treatment failures and are not reported in this manuscript. BCVA was recorded by either certified optometrists or trained ophthalmic technicians using ClearChart digital screens (Reichert, Depew, NY, USA) or a retro-illuminated Early Treatment Diabetic Retinopathy Study (ETDRS) chart placed at 4 m depending on the institution. All patients underwent 24/2 SITA Standard SAP at least on an annual basis in the post-operative period. SAP had been repeated typically within one month if unreliable.

### 2.3. Surgical Technique

Patients were prepped and draped in a usual sterile fashion. A fornix based conjunctival peritomy was performed in the superonasal quadrant in the majority of the cases with wide blunt dissection extending posteriorly. Wet-field cautery was utilized to achieve hemostasis. An orthogonal 3 × 4 mm (TF) or 4 × 4 mm (FT) partial thickness scleral flap was dissected extending through limbus into cornea. A sponge soaked in MMC was applied under the conjunctiva for 0.5–3 min. The duration and concentration of MMC application was at the discretion of the surgeon with 0.2 mg/mL for 2 min (TF) or 0.3 mg/mL for 3 min (FT), representing the most prevalent choices. The surgical area was then copiously irrigated with balanced salt solution (BSS). A paracentesis was created to establish access to the anterior chamber, the anterior chamber was entered under the scleral flap and trabeculectomy was performed utilizing a Kelly-Descemet’s punch or a surgical blade. An iridectomy was then performed with Vannas scissors. Two interrupted 10.0 Nylon adjustable sutures (TF) or three interrupted 10.0 Nylon sutures (FT) were used to secure the scleral flap. The anterior chamber was re-inflated with BSS and the suture tension was adjusted to allow slow egress of aqueous without collapse of the anterior chamber. The sutures were locked with two additional throws and subsequently rotated to bury the knots. Finally, the conjunctiva was closed with two interrupted 8.0 Vicryl sutures (Ethicon Inc., Johnson & Johnson, Somerville, NJ, USA). A 9.0 Vicryl running suture on a BV needle (TF) (Ethicon Inc., Johnson & Johnson, Somerville, NJ, USA) or an 8.0 Vicryl running suture (FT) was used to close the wings of the conjunctival incision in a watertight fashion. An additional 9.0 or 8.0 Vicryl mattress suture was passed parallel to the limbus to decrease the incidence of early aqueous leaks. The anterior chamber was formed with BSS and the incisions were examined for leaks. Patients received a subconjunctival injection of 0.4 mL dexamethasone disodium phosphate (4 mg/mL) and of 0.4 mL gentamicin sulfate (40 mg/mL) at the end of the case. Post-operative management with regards to medication selection and additional interventions was at the discretion of the surgeon and did not follow a specific protocol. In combined cases phacoemulsification was initially performed through a separate temporal clear cornea incision that was sutured with a single 10.0 Nylon suture followed by a trabeculectomy as described above. Intraocular lens selection was at the discretion of the surgeon.

### 2.4. Survival Analysis

We employed two different sets of criteria to determine successful preservation of visual function, the primary outcome measure in this study. In the most liberal analysis, failure to maintain visual function was established if a decline in Snellen visual acuity below the level of 20/200 or if a permanent decline in Snellen visual acuity by ≥3 lines occurred, or if standard automated perimetry demonstrated a decline of 3 dB or more in mean deviation. Additionally, failure occurred in case of subsequent incisional glaucoma surgery. The more strict analysis utilized the same criteria with the exception of requiring only 1 dB loss in mean deviation for failure.

Data were censored at the time of re-operation for glaucoma. We did not require reproducible visual field loss on two consecutive visits in the post-operative period, as this would artificially prolong survival due to the low frequency of testing. In addition, patients were experienced in visual field testing. Secondary to variability in length of follow-up, survival analysis (Kaplan–Meier method) was utilized to report on success. Due to non-uniform follow-up, as is the case in retrospective case series, the last observation was carried forward for survival analysis, but not more than 1 month during the first 6 months after surgery and not more than 2 months in the subsequent semesters, provided that the patient did not meet failure criteria during the next two subsequent visits at the clinic.

A separate analysis employed Tube versus Trabeculectomy (TVT) study criteria [9] to determine surgical success, the secondary outcome measure in this study, in our cohort by conventional measures (IOP) and a Kaplan–Meier curve was constructed accordingly. Briefly, failure was defined as IOP ≤ 5 mmHg or > 21 mmHg on two consecutive visits at least 3 months after surgery or not reduced by at least 20% with or without medications (qualified success). Additional criteria for failure were loss of light perception and further incisional glaucoma surgery including cyclophotocoagulation. Interventions at the slit lamp such as bleb needling, laser suture lysis, 5-fluorouracil injection or reformation of the anterior chamber did not qualify as failures both with respect to surgical success [19] and to visual function survival. Kaplan–Meier survival curves pertinent to surgical success were also constructed according to the World Glaucoma Association (WGA) recommendations [20]. In this case surgical success was defined as IOP ≤ 18 mmHg with at least 30% reduction in IOP with or without medication (qualified success) and no re-operation for glaucoma with preservation of at least light perception vision.

### 2.5. Statistical Analysis

Continuous pre- and post- operative parameters i.e., IOP, logMAR BCVA and medication requirements were compared by the student paired *t*-test provided that the data were normally distributed (Kolgomorov-Smirnov test). Kaplan–Meier survival curves were generated as stated above, both in respect of IOP reduction and preservation of visual function. We performed post hoc sample size calculations to determine the power of our study to detect differences in IOP and medication requirements before and after surgery using a readily available online sample size calculator (https://clincalc.com/stats/samplesize.aspx accessed on 18 December 2022). A sample size of 22 cases would be required to achieve a power of 90% at a 5% probability of a type I error to detect a 30% reduction in IOP. With respect to a 20% reduction in medication requirements a sample size of 18 cases would be required to achieve a power of 90% at a 5% probability of a type I error.

### 2.6. Risk Factor Analysis

We also analyzed potential risk factors for failure to maintain visual function in this cohort of patients. Therefore, we looked at various pre-, intra- and post-operative factors potentially differentiating survivors from non-survivors such as age, type of surgery (stand-alone trabeculectomy versus combined procedure), phakic status, post-op IOP and medication requirements, IOP fluctuation etc. Analysis of variance was utilized to identify factors of potential interest (NCSS, Kaysville, UT, USA). Factors that reached or approached statistical significance (*p* < 0.1) were subsequently included in a multivariate logistic regression model.

## 3. Results

Forty-three eyes of 43 patients were initially identified, all of them being Caucasian. Three eyes were excluded from subsequent analysis because of poor pre-op visual acuity (BCVA < 20/200). The mean age of the cohort was 71.1 ± 12.8 years (range: 33–87 years) and the mean follow-up duration was 23.3 ± 15.5 months with follow-up extending up to 55 months. Exfoliative glaucoma was the most prevalent diagnosis (40%), followed by primary open angle glaucoma (35%). The average pre-operative IOP was 26.5 ± 11.4 mmHg and patients were treated on average with 3.8 ± 1 IOP lowering agents before surgery. No patient was aphakic. Demographics of the cohort are summarized in Table 1.

IOP at the last follow-up visit was significantly reduced to a mean ± SD of 11.4 ± 4.0 mmHg (*p* = 1.3 × 10^−9^, student’s paired *t*-test). We also noted a significant reduction in medication requirements to the level of 0.9 ± 1.1 substances (*p* = 7.7 × 10^−14^, student’s paired *t*-test) at the last follow-up visit. Comparing average logMAR visual acuity between the pre-operative visit and the last follow-up appointment, best corrected Snellen visual acuity improved slightly from the level of 20/63 to the level of 20/50 most likely as a result of concurrent cataract surgery in 15% of patients and cataract surgery in the post-operative period in 15.4% of phakic patients at baseline (20% were pseudophakic patients at baseline). Outcome measures and data on the type of surgery performed are summarized in Table 2. We also looked at potential demographic discrepancies between the two centers Patients from Thessaloniki other than being older 85.2 ± 14.5 versus 66.9 ± 11.7 years compared to their Athenian counterparts (*p* = 0.006, student’s *t*-test) did not differ with respect to any other parameter.

We employed two different sets of criteria to determine successful preservation of visual function, the primary outcome measure in this study, as described in Methods. Kaplan–Meier curves were constructed accordingly and are presented in Figure 1 and Figure 2, respectively along with reasons for failure. Visual function was maintained in 77% of patients at two years according to the more liberal criteria and in 66% of patients at two years if the stricter criteria were utilized. The primary reason for failure was attributed to perimetric and not to visual acuity loss in both types of analyses, whereas only one patient underwent re-operation for glaucoma at 36 months.

The above primary outcome measures were achieved with overall satisfactory post-op IOP control. Utilizing TVT surgical success criteria we calculated an 89%, 85% and 72% rate of qualified success at one, two and three years respectively. A Kaplan–Meier survival curve of the qualified success is illustrated in Figure 3 along with the reasons for failure. The most common reason for surgical failure was inability to achieve at least 20% reduction in IOP, which mainly occurred in patients who started off with lower pre-operative pressures. One patient who failed because of persistent hypotony tolerated the low IOP of 4–5 mmHg quite well without a drop in visual acuity or hypotonous maculopathy and therefore was not revised. Utilizing WGA success criteria, the rate of qualified success was 85% and 67% at 1 and 3 years respectively (Figure 4).

Finally, we looked at potential risk factors for failure to maintain visual function. Despite achieving consistently low post-operative average IOPs in the low teens for the cohort (Figure 5), as evident by the standard error of measurement (SEM) at respective time-points, survivors and non-survivors differed with respect to post-op IOP and medication requirements as outlined in Figure 5 and Figure 6, respectively. In addition, analysis of variance identified other factors of potential interest such as low pre-operative IOP and phakic status. Interestingly enough, length of follow-up, combined procedures and the severity of glaucoma did not emerge as statistically significant risk factors for failure in the univariate analysis (Table 3). Factors approaching statistical significance at a level *p* < 0.1 were included in multivariate logistic regression model (Table 4). Only phakic status emerged as a potential risk factor of interest in the multivariate analysis without however reaching statistical significance (*p* = 0.088).

## 4. Discussion

We report a well characterized cohort of patients, with severe glaucoma, with reasonable visual outcomes after trabeculectomy or phaco-trabeculectomy. Surgical intervention is a viable approach for patients with severe unilateral glaucoma, with IOP not controlled on maximal medical treatment and/or VF progression.

Glaucoma progression is associated with increasing costs, both direct and indirect, according to a systematic review of the literature on cost-of-illness (COI) related to glaucoma in the US, Canada and Europe [3]. Yearly costs of treatment were 4 times higher in end-stage disease compared to mild glaucoma [21] in the US, with medication costs accounting for the majority of the financial burden over ophthalmologist visits and visual rehabilitation [22]. This was also confirmed in European studies [23,24]. In this retrospective study, standard guarded trabeculectomy with or without concurrent cataract surgery preserved visual function in a substantial proportion of eyes with severe glaucoma. This is important both for patients with markedly asymmetric disease with one unaffected eye and for monocular patients with advanced glaucoma who need to be informed on their prognosis after filtration surgery. Apart from failing to encounter wipe-out as has been previously reported [25], we demonstrated that trabeculectomy allows in at least 2/3 of patients to maintain perimetric function within 1 dB two years after surgery while being managed with fewer medications.

As expected, average IOP decreased significantly after surgery along with post-op medication requirements, which is similar to other reports in the literature. King et al. conducted a multicenter randomized controlled trial to compare trabeculectomy to medical treatment as primary treatment in advanced open angle glaucoma [26]. They examined 453 patients from 27 centers across the UK and showed that trabeculectomy managed to significantly lower IOP in the low teens, while maintaining visual acuity, visual field and quality of life during a 2-year follow-up period. They concluded that surgery was safe and more effective in producing sustainable low IOP levels compared to medications. However, patients included in this study had notably less perimetric loss (mean MD −14.91 dB) compared to our cohort. Sofi et al. retrospectively reported trabeculectomy outcomes on 60 end-stage POAG patients with VA ≤ 20/200 and high baseline IOP (mean 37.01 mmHg) after a mean follow-up of 12 months [27]. They encountered no cases of wipe-out and managed to reduce IOP significantly. Sethi et al. prospectively evaluated 20 patients with severely affected 10-2 visual fields (mean MD −29.33 dB) who underwent trabeculectomy and reported 2-year outcomes as IOP in the low teens with stability in visual acuity and perimetry [28]. We report a 85% rate of qualified surgical success at two years and 72% at 3 years by TVT criteria in this mixed cohort of severe glaucoma patients including in our analysis a wide range of glaucoma types, not limited to POAG. Several randomized clinical trials exclude such cases of refractory glaucomas [9,29]. Additionally, 15% of patients in this cohort received combined phaco-trabeculectomy which may have adversely affected surgical outcomes [30]. Furthermore, advanced disease at the time of surgical intervention may either be due to delayed diagnosis or due to a longer interval of medical treatment which may affect the state of the conjunctiva possibly influencing surgical outcomes [31]. Most cases in our cohort (n = 3) that failed by traditional IOP related criteria did so because of inability to achieve a sustainable IOP reduction of at least 20%. Two of these failures had a pre-op IOP of 13 mmHg with very thin CCT (≈440 μm) and were still progressing pre-operatively allowing for a very narrow therapeutic window.

A post hoc analysis from the Advanced Glaucoma Intervention Study showed that a sub-cohort of patients with all post-op IOPs below 17 mmHg and an average IOP of around 12.3 mmHg remained stable on average by perimetric criteria. This patient population presented with less advanced visual field loss at baseline compared to our study [32,33,34]. This is the reason why trabeculectomy is the preferred choice over less invasive techniques (i.e., MIGS) in patients with advanced glaucoma as it can achieve such low target pressures. However, other surgical approaches that achieve similar pressures, such as deep sclerectomy, could also be considered.

All eyes, regardless of the presence of glaucoma, demonstrate age-related retinal ganglion cell (RGC) loss of about 7000 RGCs per year [12]. This rate can be more substantial in patients with low remaining RGC counts and therefore even age-related loss can have a more substantial impact on visual function in patients with advanced disease It has been calculated that patients with progressive glaucoma may demonstrate a mean rate of RGC loss of about −33,000 cells/year [35]. In a retrospective study of patients with various degrees of glaucoma severity managed in a clinic with a variety of management options, the average rate of mean deviation loss on SAP was estimated at −0.45 ± 0.7 dB/year [36]. Furthermore, diagnostic tests such as SAP, may not demonstrate a linear behavior across progressive stages of the disease. In that context patients from the Ocular Hypertension Treatment Study progressed predominantly by structural criteria [37], whereas patients from the Early Manifest Glaucoma Study progressed predominantly by perimetric criteria [38]. Harwerth et al. demonstrated in experimental glaucoma in Rhesus monkeys that perimetric sensitivity in dBs bears a linear relationship to histological RGC counts expressed in dBs as well [39,40]. Utilizing a mathematical model derived from structural, perimetric and histological data it has been estimated that a 10,000 cell loss corresponds to a practically undetectable change in healthy controls with respect to global indices (mean deviation) in SAP, whereas it corresponds to a −0.71 dB change in mean deviation if there is already significant loss at the level of ~−25 dB [17]. Therefore, we may argue that whereas a 1 dB per year loss may represent the natural course of the disease without treatment in early disease [41], in patients with advanced glaucoma we should anticipate a faster decline in perimetric parameters due to age-related loss alone. Hence, a threshold of 0.5 dB/year decline may not be unreasonable in patients with advanced glaucoma and according to our data we were able to halt progression under the aforementioned level in 66% of our patients at 2 years. Successful preservation of visual function drops at 3 years in our study precipitously, because a rate of loss of −0.33 dB/year is probably unsustainable.

The selection of 2 separate perimetric thresholds for failure represents an attempt to resolve the problem of reduced specificity/increased sensitivity with more liberal criteria (i.e., decline > 3 dB) and vice versa in an event type analysis. Unfortunately, our data and frequency of testing did not permit a rate type of analysis or the utilization of dedicated glaucoma progression analysis software. The decision to utilize a wider field assessment 24° versus 10° may have contributed to fatigue in more severe glaucoma patients but was based on the requirement of one uniform tool for patient assessment, on the fact that occasionally end-stage patients may maintain a temporal island of vision and because patients with −20 to −25 dB mean deviation on SAP may have preserved visual field function beyond 10°.

Worsening of visual acuity may be attributed to cataract progression in phakic patients, to disease progression itself, to retinal or corneal pathology and to all of the above. The treating physicians have attributed decline in visual function (both acuity and/or perimetry) to glaucoma and/or progression of cataracts in all cases. However, subtle unnoticed ocular pathology may have also contributed to ocular morbidity. Unlike randomized controlled trials, this study did not exclude patients with other underlying pathologies but we have not specifically analyzed reasons for loss of visual acuity in our cohort. We selected the threshold of 3 lines decline in BCVA to remain in line with previous studies [42,43,44]. The issue of unexplained, permanent severe loss of central vision (wipe-out or snuff-out) after trabeculectomy has not been sufficiently clarified [16,25,42,43,45,46,47]. We confirmed that wipe-out in a population with advanced glaucoma most likely remains a theoretical risk, as no patient lost 3 lines of vision or more during the first 18 months after surgery. A bias due to a possible preconception that wipe-out occurs more frequently in patients with severely advanced visual field loss and therefore an underrepresentation of very advanced cases cannot be ruled out, but it should be noted that this study included only patients with a mean deviation worse than −20 dB in SAP, 44% of whom had MD worse than −28 dB and therefore represents a cohort with more terminal glaucoma cases in comparison to most of the aforementioned studies. Jampel et al. looked retrospectively at trabeculectomy outcomes and reported that 20% of patients lost ≥3 lines of visual acuity after almost 4 years of follow-up in a cohort that included all glaucoma patients older than 12 years [48]. Similarly, in the TVT study 34% of patients lost more than 2 Snellen lines of vision 5 years after surgery [9]. Both studies were not restricted to patients with advanced glaucoma. In our cohort 9% of patients lost 3 lines of vision or more at 3 years. We therefore conclude that the concern of losing visual acuity may be valid but should not influence our decision to proceed with surgery in patients with advanced glaucoma. It should be noted that patients losing visual acuity were not counterbalanced by overutilization of cataract surgery as only 4 (15.4%) underwent phacoemulsification in our study during follow-up. This is at least equivalent or lower compared to what has been reported in the Collaborative Initial Treatment Glaucoma Study (19% in 5 years) [49] and in the Singapore 5-FU Trabeculectomy Study (almost 50% in 3 years) [29].

Balekudaru et al. prospectively looked at the incidence or early visual loss (2 months after surgery) by perimetric and visual acuity criteria in patients with advanced glaucoma undergoing trabeculectomy and phaco-trabeculectomy [50]. They report a 3% incidence of 2 lines of vision or more decline, which is however mainly attributed to anticipated early post-op complications. In our study we did not look specifically at complication rates but we have no reason to believe that complication rates were any different compared to reports in the literature which estimate the total number of patients experiencing minor or major complications as high as 70% [11]. It should be noted that we did not encounter any visually devastating early or late complications such as suprachoroidal hemorrhage or endophthalmitis, which may have occurred with larger sample sizes.

Much et al. looked at patients with advanced glaucoma over an extended period (average follow-up of 8.3 years) that were managed with or without surgery at a tertiary glaucoma referral center by numerous physicians [44]. They reported a 82% survival rate in their cohort by similar criteria (patients not losing 3 dB in 10/2 SAP and not dropping below the 20/200 visual acuity level and not losing 3 lines of visual acuity). If we employ their criteria in our cohort we conclude on a 68% survival rate at 3 years, which is substantially lower. Nevertheless, their cohort did not specifically look only at patients undergoing surgery (about half underwent trabeculectomy), included predominantly black patients with higher IOP during follow-up (15.4 ± 3.2 mmHg), better initial visual acuity (mean visual acuity between 20/25 and 20/30) and less advanced visual field loss (baseline mean deviation −19.74 ± 5.60 dB). On one hand, at this level of damage, age related loss may correspond to a more modest decline in mean deviation compared to patients with more advanced disease, and on the other hand, some patients who are selected for surgery are progressing and may have therefore a more aggressive disease.

Risk factor analysis in this cohort has been inconclusive in particular due to limited power. Moreover, the regression coefficients as indicated in Table 4 demonstrate a small if any effect on visual outcomes. Nevertheless, there is a clear separation in post-op IOP and medication requirements between survivors and non-survivors based on functional criteria. For risk factor analysis the threshold of 3 dB may be more relevant with respect to visual field loss as 1 dB may occasionally fall within long-term fluctuations that are fairly common in patients with glaucoma [51]. We included in our risk factor analysis parameters that could be useful in the assessment of the risk of visual loss and not parameters over which there is no control i.e., whether patients were experiencing complications or not. We did not include in our risk analysis β-zone peripapillary atrophy or disk hemorrhages as not all of our patients had adequate documentation of such parameters. Baseline phakic status which emerged as a potentially important risk for factor for failure with respect to preservation of visual function, without however reaching statistical significance, may be attributed to the well-established observation that filtration surgery is associated with cataract progression which may also necessitate cataract surgery which in turn affects the function of the filtering bleb.

Limitations of this study include retrospective and uncontrolled nature, less than ideal frequency of perimetric testing due to practical considerations, inclusion of patients with several types of glaucoma, small sample size and therefore rather limited power for appropriate risk factor analysis and failure to assess quality of life issues which would however be difficult to interpret in patients with considerably asymmetric disease. Surgical technique was also somewhat different between the two surgeons, as it is practically impossible to protocolize this in a retrospective study. Furthermore, management decisions on escalation of treatment were at the discretion of the treating physician and not predefined and therefore non-uniform.

In conclusion, preservation of visual function is feasible in a substantial proportion of patients with fairly advanced glaucoma almost half of which had a mean deviation in SAP below −28 dB pre-operatively. Additional research is required to conclude on potential risk factors for failure to preserve visual function. As in previous studies, wipe-out was not encountered in the extended peri-operative period.

## Figures and Tables

**Figure 1 jcm-12-01639-f001:**
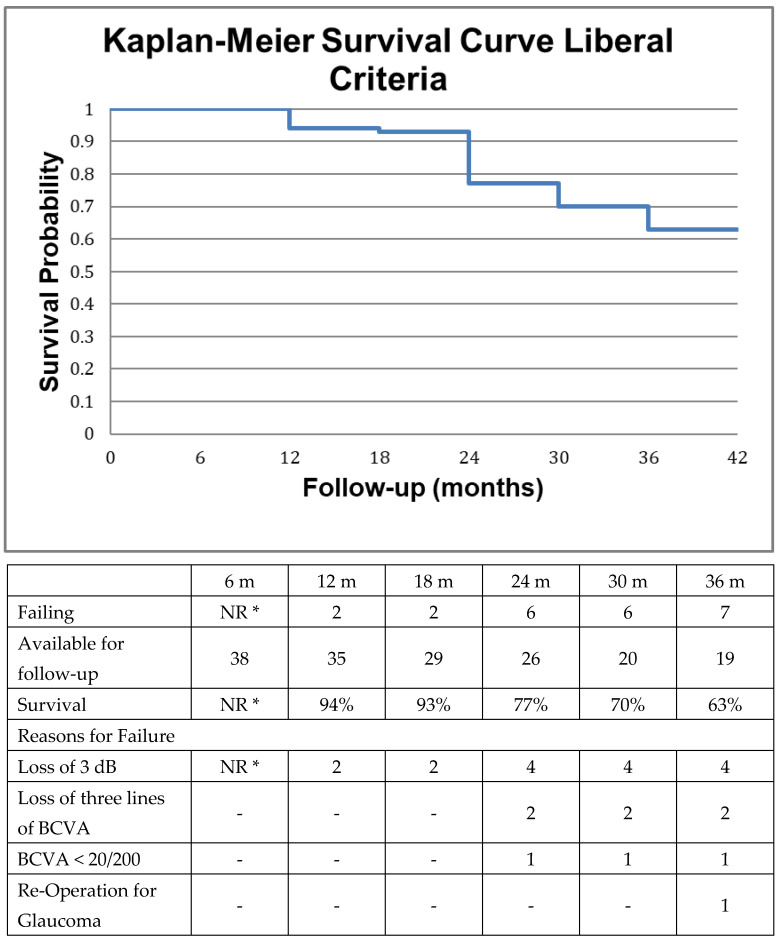
Survival of visual function according to the liberal criteria along with the reasons for failure as defined in the method section. Briefly, failure was defined as loss of 3 dBs in mean deviation in standard automated perimetry or loss of three lines in best corrected visual acuity or decline of Snellen best corrected visual acuity below the level of 20/200 or re-operation for glaucoma during the follow-up period. The individual reasons for failure may occasionally add up to more than the sum of failures because of patients failing for more than one reason. NR *: Not reported due to scarcity of VF data at 6 months.

**Figure 2 jcm-12-01639-f002:**
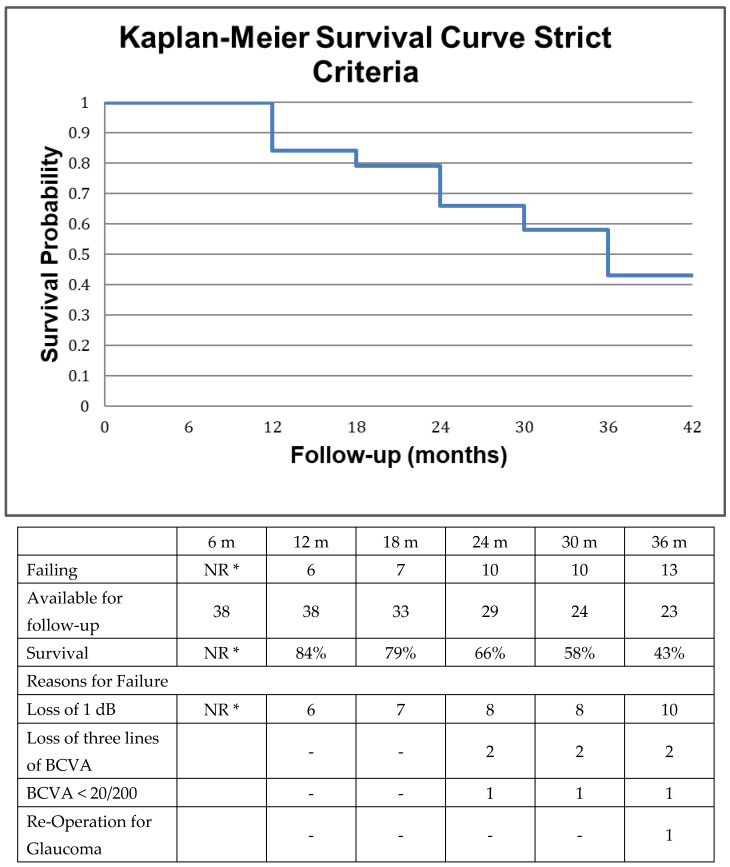
Survival of visual function according to the strict criteria along with the reasons for failure as defined in the method section. Briefly failure was defined as loss of 1 dB in mean deviation in standard automated perimetry or loss of three lines in best corrected visual acuity or decline of Snellen best corrected visual acuity below the level of 20/200 or re-operation for glaucoma during the follow-up period. The individual reasons for failure may occasionally add up to more than the sum of failures because of patients failing for more than one reason. NR *: Not reported due to scarcity of VF data at 6 months.

**Figure 3 jcm-12-01639-f003:**
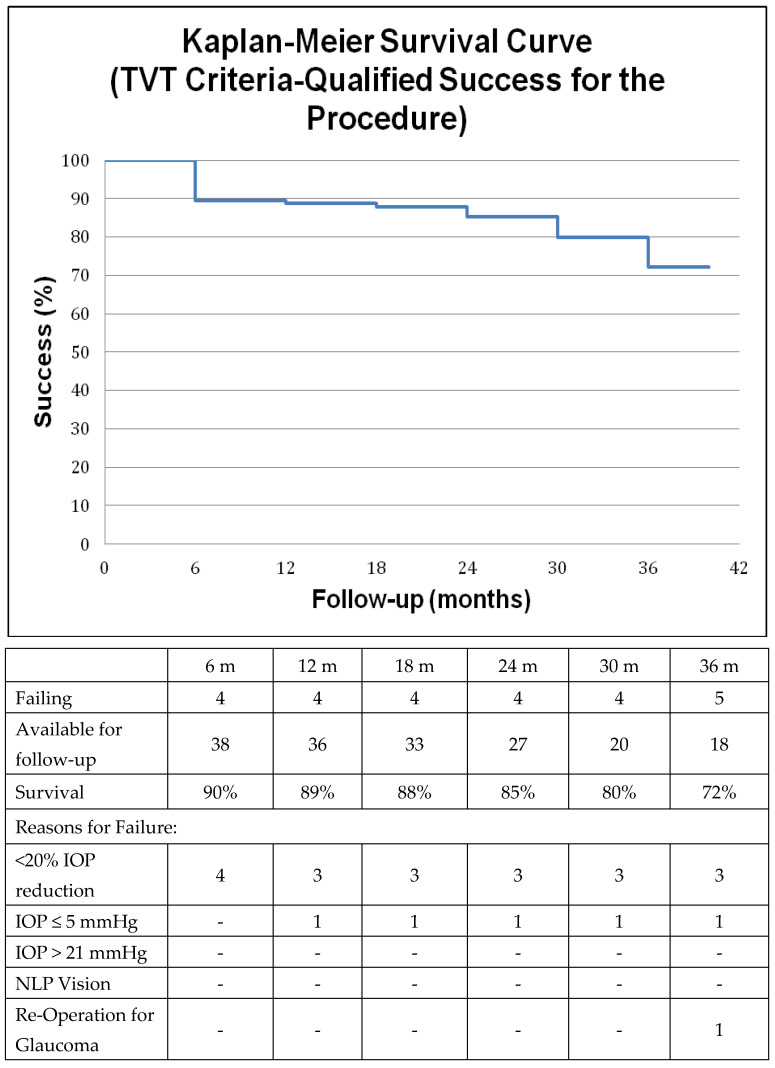
Kaplan–Meier survival curve with or without medications (qualified success) for the surgical procedure according to the Tube versus Trabeculectomy Study (briefly, failure was defined as intraocular pressure (IOP) > 21 mmHg or <6 mmHg on two consecutive visits, or IOP not reduced by at least 20%, or loss of light perception or re-operation for glaucoma).

**Figure 4 jcm-12-01639-f004:**
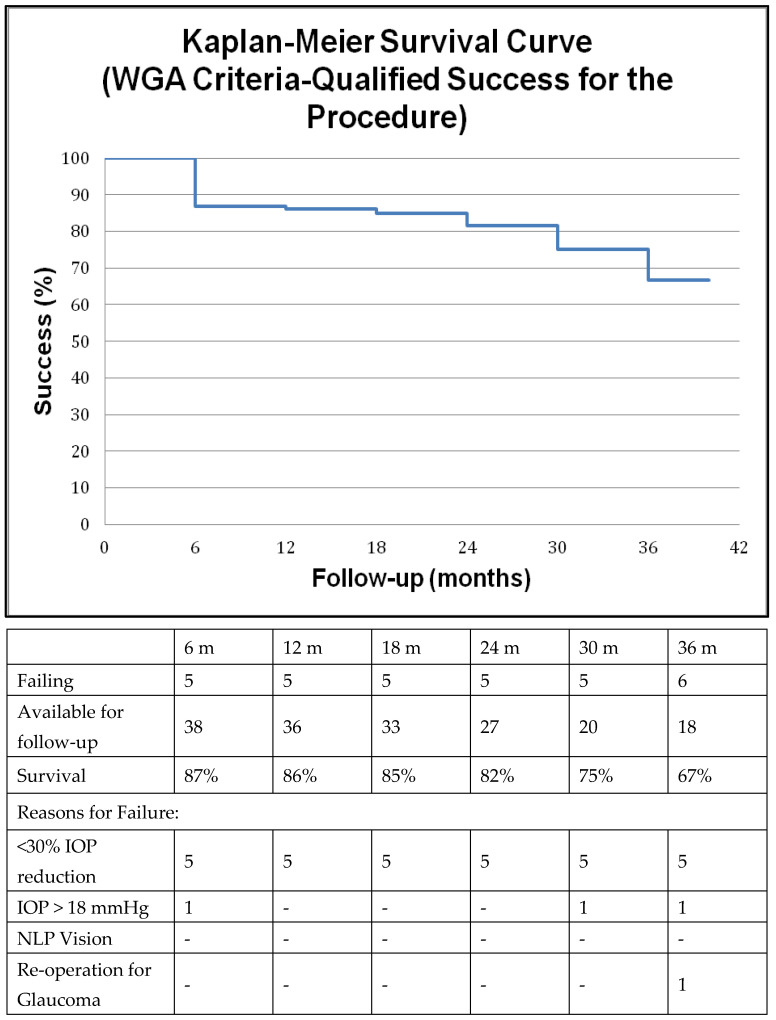
Kaplan–Meier survival curve with or without medications for the surgical procedure according to the World Glaucoma Association (briefly failure was defined as intraocular pressure (IOP) > 18 mmHg, or IOP not reduced by at least 30% or re-operation for glaucoma or loss of light perception.

**Figure 5 jcm-12-01639-f005:**
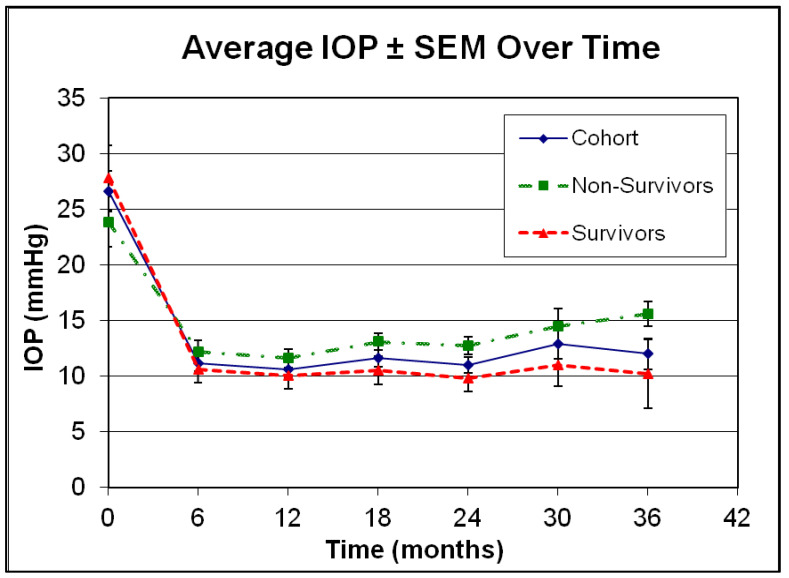
Course of average intraocular pressure (IOP) ± standard error of measurement (SEM) for all patients (cohort), for those who failed by visual function criteria (non-survivors) and for those who successfully maintained visual function (survivors), as outlined in methods. Error bars represent standard error of measurement at respective time-points. IOP differed between survivors and non-survivors significantly at 18 months (*p* = 0.04, one tail student’s *t*-test), 24 months (*p* = 0.02, one tail student’s *t*-test) and 36 months (*p* = 0.03, one tailed student’s *t*-test).

**Figure 6 jcm-12-01639-f006:**
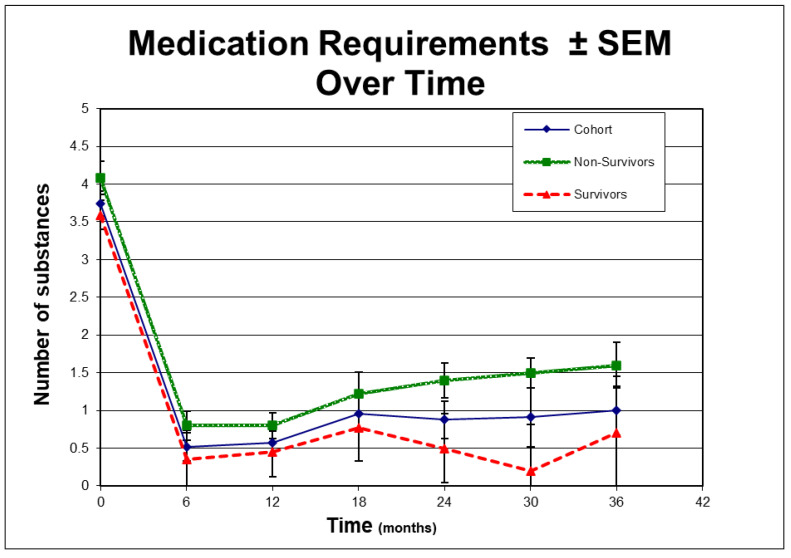
Average post-op medication requirements for all patients (cohort), for those who failed by visual function criteria (non-survivors) and for those who successfully maintained visual function (survivors) as outlined in methods. Error bars represent standard error of measurement (SEM) at respective time-points. Medication requirements differed between survivors and non-survivors significantly at 24 months (*p* = 0.03, one tail student’s *t*-test) and 30 months (*p* = 0.05, one tail student’s *t*-test).

**Table 1 jcm-12-01639-t001:** Demographic characteristics of the cohort. POAG refers to primary open angle glaucoma and includes low tension glaucoma cases, XFG refers to exfoliative glaucoma, “*other” includes mixed mechanism glaucoma (three cases), angle closure glaucoma (two cases), pigmentary glaucoma (one case), uveitic glaucoma (one case), juvenile open angle glaucoma (one case), steroid induced glaucoma (one case) and glaucoma due to increase in episcleral venous pressure with blood in Schlemm’s canal (one case). IOP refers to intraocular pressure; SAP refers to standard automated perimetry.

Demographic Characteristics	
No. of Eyes	40
Average Age ± SD, range (Years)	71.1 ± 12.8, 33–87
Gender N, % (Male)	22, 55%
Race N, % (Caucasian)	40, 100%
Average Length of Follow-up ± SD, range (months)	23.3 ± 15.5, 1–55
Average Pre-op IOP ± SD (mmHg)	26.5 ± 11.4
Average Pre-op Medications ± SD (substances)	3.8 ± 1
Average Central Corneal Thickness ± SD (μm)	524.5 ± 41.2
Average Mean Deviation on SAP ± SD, range (dB)	−26.3 ± 4.1, −20.1 to −33.29
−20.01 to −24.00	39%
−24.01 to −28.00	17%
>28.01	44%
Eyes with Split Fixation N, %	32 (80%)
Eyes with Previous Laser Glaucoma Surgery N, %	2 (5%)
Pseudophakia N, %	8 (20%)
Diagnosis N, %	
Primary Open Angle Glaucoma (POAG)	14 (35%)
Exfoliative Glaucoma (XFG)	16 (40%)
*Other:	10 (25%)

**Table 2 jcm-12-01639-t002:** Pre-operative characteristics and operative outcomes of the cohort at the last available follow-up visit. Post-op data are in reference to the last available visit and are censored at the time of re-operation for glaucoma. *p*-values refer to statistical comparison by the paired *t*-test in all instances.

Outcomes and Operative Characteristics		*p*-Value
Average Pre-op IOP ± SD (mmHg)	26.5 ± 11.4	1.3 × 10^−9^
Average Post-op IOP ± SD (mmHg)	11.4 ± 4.0	
Average Pre-op Medications ± SD (substances)	3.7 ± 1.0	7.7 × 10^−14^
Average Post-op Medications ± SD (substances)	0.9 ± 1.1	
Average Pre-op logMAR Visual Acuity ± SD	0.5 ± 0.4	0.9
Average Post-op logMAR Visual Acuity ± SD	0.4 ± 0.5	
Number of Combined Cases N, % (+ uncomplicated phacoemulsification	6, 15%	

**Table 3 jcm-12-01639-t003:** Univariate risk factor analysis of various pre-, intra- and post-op parameters possibly affecting survival of visual function (not bleb survival) in this cohort of patients with advanced glaucoma undergoing standard guarded trabeculectomy or phaco-trabeculectomy.

Independent Variables	Survival of Visual Function Strict Criteria Risk Factor Analysis (ANOVA)	Survival of Visual Function Liberal Criteria Risk Factor Analysis (ANOVA)
Pre-op Characteristics		
	*p*-value	*p*-value
Age per decade	0.22	0.44
Gender	0.21	0.16
Diagnosis other than POAG	0.55	0.52
Mean Deviation on SAP (per 4 dB)	0.44	0.86
IOP (mmHg)	0.032 *	0.38
Central Corneal Thickness (per 40 μm)	0.61	0.27
Glaucoma Medication Requirements	0.71	0.97
Phakic Status	0.17	0.068 #
logMAR BCVA	0.71	0.63
Operative Characteristics		
Surgeon	0.58	0.14
Combined Cases	0.46	0.25
Post-operative Characteristics		
IOP at 3 months (mmHg)	0.19	0.36
IOP at 24 months (mmHg)	0.045 *	<0.0001 *
IOP fluctuations (mmHg)	0.74	0.82
Glaucoma Medication Requirements at 24 months	0.08 #	0.87
Length of Follow-up (per year)	0.50	0.77

* denotes parameters with statistically significant impact. # denotes parameters approaching but not achieving statistical significance.

**Table 4 jcm-12-01639-t004:** Multivariate risk factor analysis. Two models of logistic regression including independent variables with *p* < 0.1 in the univariate analysis.

Independent Variables	Survival of Visual Function Strict Criteria Risk Factor Analysis (Logistic Regression)			Survival of Visual Function Liberal Criteria Risk Factor Analysis (Logistic Regression)		
Model 1						
	*p*-value	Regression Coefficient	Standard Error	*p*-value	Regression Coefficient	Standard Error
Pre-op IOP (mmHg)	0.51	2.7 × 10^−2^	4 × 10^−2^	0.79	1.2 × 10^−2^	4.7 × 10^−2^
Phakic Status	0.21	1.1	0.9	0.097 #	1.6	1
IOP at 24 months (mmHg)	0.92	−1.5 × 10^−2^	0.2	0.45	−0.14	0.2
Glaucoma Medication Requirements at 24 months	0.36	−0.4	0.4	0.53	0.3	0.5
Model 2						
Phakic Status	0.18	1.1	0.8	0.088 #	1.6	1
IOP at 24 months (mmHg)	0.48	−8 × 10^−2^	0.1	0.63	−7 × 10^−2^	0.1

# denotes parameters approaching but not achieving statistical significance.

## Data Availability

The principal investigators, Thodoris Filippopoulos and Fotis Topouzis, had full access to all the data in the study and take responsibility for the integrity of the data and the accuracy of the data analysis. The data presented in this study are available on request from the corresponding author.

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
