# Peer review of "Survival of Visual Function in Patients with Advanced Glaucoma after Standard Guarded Trabeculectomy with MMC"

_jcm, 2023, doi:10.3390/jcm12041639_

Round 1

Reviewer 1 Report

The authors present a retrospective study to assess the effect of glaucoma filtering surgery on visual field and visual acuity in patients with advanced glaucoma.

Although the authors state among the study's limitations, among others, the small sample size, no estimation of the sample size needed to conclude has been made prior to the study. The authors should include this in the material and methods section and also in the discussion.

On the other hand, special emphasis should be done on the different surgical technique performed by the two surgeons and especially on the different postoperative treatment which, although it was different between surgeons, as far as I understood, was not protocolized either.

Regarding the Results section, the survival analysis is extensive, applying different criteria. I feel that if the WGA success criteria had been considered, it would have been enough.

Analysis of the factors that could influence the loss of visual acuity or visual field was carried out using univariate and multivariate analysis. Beta slope and 95% confidence interval should be included in table 4. I would be grateful if the authors could add them so that the effect of each parameter on the dependent variable could be better identified.

Author Response

We would like to thank our colleague for the very thorough review of our manuscript. Please find enclosed our response to your insightful comments. 

The authors present a retrospective study to assess the effect of glaucoma filtering surgery on visual field and visual acuity in patients with advanced glaucoma.

Although the authors state among the study's limitations, among others, the small sample size, no estimation of the sample size needed to conclude has been made prior to the study. The authors should include this in the material and methods section and also in the discussion. This is indeed a very serious omission on our behalf. We have therefore performed post-hoc sample size calculations and have included these under statistical analysis in the method section: "We performed post-hoc sample size calculations to determine the power of our study to detect differences in IOP and medication requirements before and after surgery using a readily available on line sample size calculator (https://clincalc.com/stats/samplesize.aspx). A sample size of 22 cases would be required to achieve a power of 90% at a 5% probability of a type I error to detect a 30% reduction in IOP. With respect to a 20% reduction in medication requirements a sample size of 18 cases would be required to achieve a power of 90% at a 5% probability of a type I error."

On the other hand, special emphasis should be placed on the different surgical technique performed by the two surgeons and especially on the different postoperative treatment which, although it was different between surgeons, as far as I understood, was not protocolized either. This is certainly a limitation of our study and it has been added as such under limitations in the discussion section of this paper as it is impossible to protocolize surgical technique in a retrospective study. Surgical technique however was not that different between the two surgeons in this study. Post-operative management decisions may well be different as they were at the discretion of the treating physicians. This Is a limitation of even well-designed prospective studies. For example in the TVT the decision to proceed with subsequent de novo surgery in surgical failures was at the discretion of the treating physicians and therefore the investigators compared the intraocular pressures at which this occurred in the two cohorts. 

Regarding the Results section, the survival analysis is extensive, applying different criteria. I feel that if the WGA success criteria had been considered, it would have been enough. We can certainly present one or the other (TVT vs. WGA) surgical success criteria in the main body of the manuscript and present the other one as supplementary material. We have been criticized in a previous submission for leaving the TVT criteria out as reviewers felt that readers who are not glaucoma specialists may be more familiar with the TVT criteria of surgical success or we could present both data in one graph. 

Analysis of the factors that could influence the loss of visual acuity or visual field was carried out using univariate and multivariate analysis. Beta slope and 95% confidence interval should be included in table 4. I would be grateful if the authors could add them so that the effect of each parameter on the dependent variable could be better identified. This has been rectified accordingly. 

Reviewer 2 Report

The study aimed to determine the effect of trabeculectomy or combined phaco-trabeculectomy on the visual function of advanced glaucoma patients, which proved Trabeculectomy and/or phaco-trabeculectomy is associated with preservation of visual function in patients with uncontrolled advanced glaucoma. The article evaluated the success rate of surgery from the perspective of visual function——a novel perspective.

1.  The description in the methods was like a prospective study rather than a retrospective study. It is recommended to refer to the writing standards of observational studies.

2.  The article mentions thatFollow-up data were typically recorded on day one, day 3, week one, week 2 and 1, 3, 6, 12 months postoperatively and every 6 months thereafter and included IOP, glaucoma medication requirements, BCVA, early and late complications related to surgery”, we were unable to find results at day one, day 3, week one, week 2 after surgery from the results section.

3.  Survival analysis and risk factor analysis could be included in the statistical analysis. In addition, the sample size for the risk-factor analysis was too small to support a multivariate analysis. More important influencing factors can be selected for univariate analysis, or risk factors can be deleted from the statistical analysis.

4.  The results about primary outcomes,secondary outcomes and surgical complications should be discussed more specifically in the discussion section of this paper.

Author Response

We would like to thank our colleague for the very thorough review of our manuscript. Please find enclosed our response to your insightful comments. 

  1. The description in the methods was like a prospective study rather than a retrospective study. It is recommended to refer to the writing standards of observational studies. In the abstract and in the main body of the manuscript it is stated 5 times that this is a retrospective study. Furthermore, in the methods section it is stated that due to the non-uniformity of follow-up, the last observation was carried forward but not more than one month during the first 6 months of the study and not more than 2 months thereafter provided that patients did not fail during subsequent visits. We understand the confusion the use of the simple past tense may have generated and also changed phrasing and the tense into the past perfect tense in the section under clinical data to make the nature of the study more obvious i.e. line 101 …for this retrospective study… or line 110 …had been typically recorded… or line 117 …had been at the discretion of the treating physician…
  2. The article mentions that “Follow-up data were typically recorded on day one, day 3, week one, week 2 and 1, 3, 6, 12 months postoperatively and every 6 months thereafter and included IOP, glaucoma medication requirements, BCVA, early and late complications related to surgery”, we were unable to find results at day one, day 3, week one, week 2 after surgery from the results section. Thank you for this comment. We decided to omit IOP results during the first two weeks after surgery as they may not contribute significantly to long or intermediate term visual outcomes. Trabeculectomy is a titratable procedure and early on efforts are being made to approach the low target pressures that are required in such advanced glaucoma cases by laser suture lysis or selective removal of releasable sutures. One metric that may appear pertinent to our study are early IOP spikes (defined as ΔIOP>10mmHg compared to baseline or IOP>30mmHg) but this metric did not approach significance in our risk factor analysis probably due to low power and small sample size. Of note we encountered one case of each in the early post-operative period in this cohort.  We would be happy to address this issue by adding respective timepoints to figures 5 & 6 respectively if you think this is of great importance.
  3. Survival analysis and risk factor analysis could be included in the statistical analysis. There is in this manuscript in the methods section a paragraph on Kaplan Meier survival curves both with respect to surgical success and successful preservation of visual function.
  4. In addition, the sample size for the risk-factor analysis was too small to support a multivariate analysis. More important influencing factors can be selected for univariate analysis, or risk factors can be deleted from the statistical analysis. We have added post hoc sample size calculations on the comparison of IOP and medication requirements before and after surgery. This has been a comment by one of the other reviewers as well. With respect to the evaluation of risk factors for failure to preserve visual function it is certainly true that our sample size is rather small to be able to detect subtle meaningful associations. To prevent overinterpretation of the available data we have added on table 4 the respective regression coefficients which are rather small and the respective standard errors of measurements of our analysis. We have also added in the discussion right after: “Risk factor analysis in this cohort has been inconclusive in particular due to limited power.” the following sentence (line 495): “Moreover, the regression coefficients as indicated in table 4 demonstrate a small if any effect on visual outcomes.”
  5. The results about primary outcomes, secondary outcomes and surgical complications should be discussed more specifically in the discussion section of this paper. This is a small retrospective study and there are larger prospective studies on reporting surgical complication of filtration surgery. Our goal was to investigate if filtration surgery can have a reasonable effect on preservation of visual function in patients with very advanced disease. In other words, is it worthwhile to operate on such patients? The only reason why we report IOP related outcomes is to make clear that surgical success is similar to what is expected in filtration surgery. We have also been criticized that the manuscript is already long enough. Adding one more table on complications would make it even longer. We also plan on reporting complications in a subsequent paper where we compare surgical complications in patients with advanced glaucoma in comparison to patients with less advanced disease. We have rephrased certain sentences in the methods and results section to directly reflect what the primary and secondary outcome measures have been. i.e. line 175: “…surgical success, the secondary outcome measure in this study,…” and line 157: “ …preservation of visual function, the primary outcome measure in this study…” and line 246 “ … visual function, the primary outcome measure in this study…”

Reviewer 3 Report

Authors investigated visual function preservation in patients with advanced glaucoma following trabeculectomy or combined phaco-trabeculectomy in a retrospective, non-comparative, interventional study design. Primary outcome was survival of visual function: visual acuity and VF parameters using 5 criteria. Secondary outcome was qualified surgical success using 2 sets of criteria.

The manuscript is comprehensively written, reads well and reflects real world situation.

I would suggest adding to the section “Statistical analysis” also analyses used in the study (survival analysis….)

The main confounder is including patients with both trabeculectomy and combined phaco-trabeculectomy (15%) or later undergoing cataract surgery (15%) as the latter influences VA outcome which was one of the visual function preservation criteria. In addition, patients with advanced glaucoma have increased variation in VF and performing post-op VF testing on an annual basis may not be a reliable indicator to assess visual function preservation. This was also mentioned in the Discussion.

Age-related macular present in an eye with advanced glaucoma affects visual acuity and its change during follow-up. Was AMD degeneration among exclusion criteria?

Specific comments:

Tables 3 and 4: typing error, substitute “@” with “at”

Page 14; line 382: “This rate can be more substantial in patients with low remaining RGC counts.” Probably it is meant that loss of RGC per year (even age-related) has greater impact on visual function in patients with advanced glaucoma?

Author Response

We would like to thank our colleague for the comprehensive review. Please find enclosed our response to your insightful comments.

  1. I would suggest adding to the section “Statistical analysis” also analyses used in the study (survival analysis….) This comment has been addressed.
  2. The main confounder is including patients with both trabeculectomy and combined phaco-trabeculectomy (15%) or later undergoing cataract surgery (15%) as the latter influences VA outcome which was one of the visual function preservation criteria. In addition, patients with advanced glaucoma have increased variation in VF and performing post-op VF testing on an annual basis may not be a reliable indicator to assess visual function preservation. This was also mentioned in the Discussion. This is true. However, it is known that filtration surgery is associated with worsening cataracts. Excluding patients that later on developed visually significant cataracts and have been operated for worsening cataracts would also bias our results and illustrate a situation that does not reflect clinical practice where depending on the severity of cataracts clinicians decide either on performing trabeculectomy alone or combined phacotrabeculectomy. The aim of this study has been to investigate if filtration surgery can delay the inevitable in very advanced disease which may be irrelevant in patients with bilateral disease but is very important in patients with markedly asymmetric disease. Visual test variability is also an established fact especially in patients with advanced disease. This is a retrospective study on patients with advanced disease often travelling from further away to receive care and often undergoing additional perimetric testing with various devices at the offices of referring physicians. We could not perform more frequent perimetric testing due to practical considerations. This fact is mentioned in the discussion under limitations (line 509)" the less than ideal frequency of perimetric testing due to practical considerations.."

Age-related macular present in an eye with advanced glaucoma affects visual acuity and its change during follow-up. Was AMD degeneration among exclusion criteria?

Patients with non-exudative age related macular degeneration were not excluded from our analysis. Certainly progression of unrelated ocular pathology can contribute to visual loss reflected in visual acuity or perimetry. In the discussion we have stated: "Worsening of visual acuity may be attributed to cataract progression in phakic patients, to disease progression itself, to retinal or corneal pathology and to all of the above." We added right after this sentence the following (line 440): "The treating physicians have attributed decline in visual function (both acuity and/or perimetry) to glaucoma and/or progression of cataracts in all cases. However, subtle unnoticed ocular pathology may have also contributed to ocular morbidity. " No patients developed exudative macular degeneration during follow-up. 

Specific comments:

Tables 3 and 4: typing error, substitute “@” with “at”It has been addressed accordingly. 

Page 14; line 382: “This rate can be more substantial in patients with low remaining RGC counts.” Probably it is meant that loss of RGC per year (even age-related) has greater impact on visual function in patients with advanced glaucoma? You are right. We added for clarity (line 403): and therefore even age related loss can have a more substantial impact on visual function in patients with advanced disease. 

Reviewer 4 Report

The work addresses an important issue, the surgical approach of advanced glaucoma.

Introduction: "Standard guarded trabeculectomy with antimetabolites remains the most reasonable surgical option for patients with severe glaucoma". I'm not sure trabeculectomy is the most reasonable, it is probably still the most used according with surveys. In addition someone could question that less invasive technique can be more reasonable in advanced cases. I would consider to rephrase "reasonable" and to discuss this part. 

Abstract "Trabeculectomy and/or phaco-trabeculectomy is associated with preservation of visual function in patients with uncontrolled advanced glaucoma". Looking at the data, fig 1 shows a loss of 3dBs in 37% of patients (36 mts) and fig 2 a loss of 1dB in 57% of patients (36 mts). I'm not sure these data are consistent with the sentence "preservation of visual function", I suggest to rephrase in term of more caution and to discuss. 

Safety data are lacking and a detailed complication report is missing, this is an important point to be addressed. 

How many patients underwent combined phaco/trabe? It is possible to compare the outcome of trabe alone and phaco/trabe (see for reference Sacchi et al. EJO 2022)?

Patient underwent surgeries between January 2010 and December 2011, it is considerable time ago. Can authors explain the reason of this gap? 

Deep sclerectomy and less invasive glaucoma surgery are also valuable option for the management of advanced glaucoma, I would mention it in discussion. 

Author Response

We would like to thank our colleague for the very thorough review of our manuscript. Please find enclosed our response to your insightful comments. 

The work addresses an important issue, the surgical approach of advanced glaucoma.

Introduction: "Standard guarded trabeculectomy with antimetabolites remains the most reasonable surgical option for patients with severe glaucoma". I'm not sure trabeculectomy is the most reasonable, it is probably still the most used according with surveys. In addition someone could question that less invasive technique can be more reasonable in advanced cases. I would consider to rephrase "reasonable" and to discuss this part.  We have rephrased the above phrase which reads now as: " the most commonly employed"  We have added in the discussion the following phrase right after referencing the AGIS (line 397)."This is the reason why trabeculectomy is the preferred choice over less invasive techniques (i.e. MIGS) in patients with advanced glaucoma as it can achieve such low target pressures. However, other surgical approaches that achieve similar pressures, such as deep sclerectomy, could also be considered. " 

Abstract "Trabeculectomy and/or phaco-trabeculectomy is associated with preservation of visual function in patients with uncontrolled advanced glaucoma". Looking at the data, fig 1 shows a loss of 3dBs in 37% of patients (36 mts) and fig 2 a loss of 1dB in 57% of patients (36 mts). I'm not sure these data are consistent with the sentence "preservation of visual function", I suggest to rephrase in term of more caution and to discuss. We have revised the abstract accordingly which reads now as: " ...trabeculectomy and/or phacotrabeculectomy is associated with meaningful visual outcomes..." to better reflect our results. With respect to the outcomes themselves. Losing 1 dB per year (-0.33dB/year may be reasonable but obviously not sufficient). There are a number of studies  (i.e. DeMoraes et al) that report even faster rates depending on the type of glaucoma on treated patients in real world clinical settings. Losing one 1dB per year is fast indeed but someone may need to consider as we have  analyzed in the discussion the non-linear behavior of perimetry as a  diagnostic test in relation retinal ganglion cell loss. The message this manuscript attempts to convey is not that we are doing a terrific job with the weapons that we have in our hands but that filtration surgery can delay blindness in a considerable proportion of patients with very advanced disease. 

Safety data are lacking and a detailed complication report is missing, this is an important point to be addressed. The safety data of this cohort are a subject of another manuscript that will be submitted in close succession. Including such data would make an already long manuscript even longer. This has been a point of criticism of our work previously when we submitted an earlier version elsewhere that included a table of early and late complications along with the relevant discussion points. Complications in this cohort have been not substantial different compared to what has been reported in the literature. The most interesting part of complication analysis is to what degree specific complications i.e. IOP spikes affect visual outcomes. Our endpoints however are few to reach meaningful conclusions in this regard. 

How many patients underwent combined phaco/trabe? It is possible to compare the outcome of trabe alone and phaco/trabe (see for reference Sacchi et al. EJO 2022)? It is stated in the result section that 15% of patients received combined phacotrabeculectomy. Thank for bringing this important study to our attention (Sacchi M et al. Eur J Ophthalmol 2022 Jan;32:327-335) We added this reference in the discussion after the sentence: " Additionally, 15% of patients in this cohort received combined phaco-trabeculectomy which may have adversely  affected surgical outcomes. " (line 385)

Patient underwent surgeries between January 2010 and December 2011, it is considerable time ago. Can authors explain the reason of this gap? This paper has been conceptualised in 2016. This is why the longest follow-up in this cohort of patients is 55 months. Then it was presented as a poster at the American Academy of Ophthalmology in Chicago (end of 2016). Once presented there, the journal Ophthalmology retains the right of first refusal. It received positive reviews, revisions were  requested and then it was rejected. Unfortunately, the turnover in some journals still remains long enough.

Deep sclerectomy and less invasive glaucoma surgery are also valuable option for the management of advanced glaucoma, I would mention it in discussion. This has been addressed. Please refer to your first comment. 

Reviewer 5 Report

-         Congratulations to the authors for this article.

-         The manuscript is appropriately referenced and authors presented sufficient data with appropriate tables and figures and the article is easy to read and logically structured.

   however, the design had wide range in inclusion criteria for the patients included in the study

 Many unrelated confounding factors are present in selection of the patients, mechanism of glaucoma for example, combining juvenile glaucoma with angle closure and open angle would bias the vision results as usually those with chronic OAG will have reduced vision compared to others.

   The authors quoted tube versus trab study as a reference in the analysis, however they did not exclude aphakic patients. Again an important confounding factor. TVT study had defined the failure as less than 5 mmHg not 6 (incorrectly written in the legend of figure 3).

  In the analysis of visual fields, including mean deviation as a sole parameter in interpreting visual field could be misleading, specially that a proportion of the patients had cataract which could reduce the mean deviation as it is affected by any cause that induce generalized reduction of retinal sensitivity. Pattern standard deviation should have been included in addition to the mean deviation.

  The study is lacking an important analysis test which we rely on in every day practice in the recent era, optical coherence tomography analysis of the optic nerve and ganglion cell complex in the macular area.

 I would like to know reasons for choosing one tailed T test in comparing IOP and medication between pre and post operative patients, as it can overestimate the significance.

Author Response

We would like to thank our colleague for the thoughtful review 

Please find enclosed our responses to your insightful comments.

The manuscript is appropriately referenced and authors presented sufficient data with appropriate tables and figures and the article is easy to read and logically structured.

   however, the design had wide range in inclusion criteria for the patients included in the study

 Many unrelated confounding factors are present in selection of the patients, mechanism of glaucoma for example, combining juvenile glaucoma with angle closure and open angle would bias the vision results as usually those with chronic OAG will have reduced vision compared to others. This is very true. We would certainly like to limit out study to patients with advanced primary open angle or exfoliative glaucoma. There were however just not enough surgical cases with so advanced disease in our practices. We have included your comment under limitations In the discussion session.

   The authors quoted tube versus trab study as a reference in the analysis, however they did not exclude aphakic patients. Again an important confounding factor. TVT study had defined the failure as less than 5 mmHg not 6 (incorrectly written in the legend of figure 3). Your are right. We did not specifically exclude aphakic patients but no patient was aphakic. We edited table 1 accordingly and we added in the result section that no patient was aphakic. The TVT study  defines failure as IOP less or equal than 5mmHg on two consecutive visits. We employ the same criterion but we have phrased it as IOP<6mmHg which is essentially the same. 

  In the analysis of visual fields, including mean deviation as a sole parameter in interpreting visual field could be misleading, specially that a proportion of the patients had cataract which could reduce the mean deviation as it is affected by any cause that induce generalized reduction of retinal sensitivity. Pattern standard deviation should have been included in addition to the mean deviation. Pattern standard deviation is indeed a very important parameter in perimetric testing. It is in our understanding that it represents a measure of the focal nature of the disease. As the disease progresses in early stages someone may encounter initial increase in PSD values but as the perimetric loss becomes more uniform in more advanced stages someone may encounter improved values despite progression of the disease. Therefore, we feel that mean deviation is more representative of what is happening in the particular cohort of patients. 

  The study is lacking an important analysis test which we rely on in every day practice in the recent era, optical coherence tomography analysis of the optic nerve and ganglion cell complex in the macular area. We decided against using structural parameters in this study for a number of reasons. Is some instances OCT imaging was not available or of poor quality due to cataract or inability to fixate. Furthermore, at this stage of disease structural parameters may demonstrate a floor effect beyond which any further decline is not anticipated. 

 I would like to know reasons for choosing one tailed T test in comparing IOP and medication between pre and post operative patients, as it can overestimate the significance. You are absolutely right that the one-tailed t-test is a more liberal statistical approach. On the other hand patients are operated to achieve lower pressures compared to before surgery and if this is not the case they constitute surgical failures and they are being re-operated. In such case they are censored from any subsequent analysis. 

Round 2

Reviewer 4 Report

The co-authors addressed effectively the points raised by the reviewers, the paper has been improved and is suitable for publication.